# Remote Wearable Neuroimaging Devices for Health Monitoring and Neurophenotyping: A Scoping Review

**DOI:** 10.3390/biomimetics9040237

**Published:** 2024-04-16

**Authors:** Mohamed Emish, Sean D. Young

**Affiliations:** 1Department of Informatics, University of California, Irvine, CA 92697-3100, USA; syoung5@hs.uci.edu; 2Department of Emergency Medicine, University of California, Irvine, CA 92697-3100, USA

**Keywords:** neuroimaging, remote health monitoring, electroencephalogram (EEG), wearable brain scanning, neuro-informatics, digital phenotyping, remote sleep monitoring, neurological disorders, mental health, chronic pain

## Abstract

Digital health tracking is a source of valuable insights for public health research and consumer health technology. The brain is the most complex organ, containing information about psychophysical and physiological biomarkers that correlate with health. Specifically, recent developments in electroencephalogram (EEG), functional near-infra-red spectroscopy (fNIRS), and photoplethysmography (PPG) technologies have allowed the development of devices that can remotely monitor changes in brain activity. The inclusion criteria for the papers in this review encompassed studies on self-applied, remote, non-invasive neuroimaging techniques (EEG, fNIRS, or PPG) within healthcare applications. A total of 23 papers were reviewed, comprising 17 on using EEGs for remote monitoring and 6 on neurofeedback interventions, while no papers were found related to fNIRS and PPG. This review reveals that previous studies have leveraged mobile EEG devices for remote monitoring across the mental health, neurological, and sleep domains, as well as for delivering neurofeedback interventions. With headsets and ear-EEG devices being the most common, studies found mobile devices feasible for implementation in study protocols while providing reliable signal quality. Moderate to substantial agreement overall between remote and clinical-grade EEGs was found using statistical tests. The results highlight the promise of portable brain-imaging devices with regard to continuously evaluating patients in natural settings, though further validation and usability enhancements are needed as this technology develops.

## 1. Introduction

Digital Phenotyping (DP) is a field in which digital tools are employed to collect data on individuals’ health and behavioral patterns, providing insights into human well-being [1]. At the heart of DP is passive data collection, wherein sensors and software in personal devices such as smartphones and wearables automatically record various health-related metrics without intervening in the users’ lives [2]. This approach allows for the continuous and real-time monitoring of health indicators, thereby generating a comprehensive view of an individual’s health status [3]. The data obtained through passive collection are instrumental in personalized public health, where they are used to design health interventions and programs tailored to individual profiles [4]. DP plays a critical role in the development of personalized, data-driven health strategies, aiming to improve preventive care and health promotion at both the individual and community levels [5].

Emerging technologies in DP are enabling the monitoring of brain activity, providing valuable insights into cognitive functions, emotional states, and neurological conditions [6,7]. By integrating brain activity data with other digital biomarkers, researchers aim to develop comprehensive neurological profiles that can inform personalized interventions and treatments for conditions such as depression, anxiety, and neurodegenerative disorders. Ecological momentary assessment (EMA), a research methodology that involves observing subjects in their natural environment [8], addresses the limitations of traditional neuroimaging techniques, which often lack real-world applicability [9]. Traditional approaches often involve simplified, decontextualized tasks that fail to capture the complexities of natural human behaviors and environments [10]. By integrating technologies like wearable neuroimaging devices into everyday settings, researchers can gather more ecologically valid data. This is crucial for understanding the nuanced interplay between brain activity, behavior, and environment [11]. This approach not only enhances the relevance of research findings for real-life situations but also opens new avenues for monitoring and intervention in healthcare and research [12].

Electroencephalograms (EEGs), functional near-infra-red spectroscopy (fNIRS), and photoplethysmography (PPG) are non-invasive neuroimaging modalities well-suited for continuous brain monitoring outside traditional lab settings due to their portability, cost-effectiveness, and minimal interference in a patient’s life [13]. An EEG measures the electrical activity of the brain by recording the voltage fluctuations resulting from ionic current flows within the neurons of the brain [14]. It captures the electrical activity of the brain at various frequency bands, where different frequencies serve as biomarkers of different health conditions. For example, alpha waves, which are in a relatively high frequency range (8–12 Hz), are associated with relaxation and calmness. In contrast, beta waves, which are lower in frequency (with a range of 12–30 Hz), are associated with highly demanding mental activity such as analytical thought and anxiety [15]. The Muse headband [16], Emotiv EPOC+ [17], and NeuroSky MindWave [18] are examples of commercial EEG headsets. In fNIRS, brain activity is measured by monitoring changes in blood oxygenation and blood volume in the cortex using near-infrared light [19]. This provides insights into which areas of the brain are more active, reflecting increased neuronal activity and metabolism. Examples of commercial fNIRS wearable devices include Mendi [20] and Kernel [21]. PPG is an optical technique involving the measurement of changes in blood volume in the microvascular bed of tissue [22]. While not a direct measure of brain activity, PPG can be used alongside an EEG or fNIRS to investigate the interaction between physiological markers related to brain function, such as heart rate variability [23].

Wearable neuroimaging devices represent an opportunity for researchers to monitor brain activity and use it as a biomarker for various health conditions. Traditional EEGs rely on multiple electrodes and conductive gels applied to the scalp in a controlled setting by trained staff [24]. fNIRS also requires careful positioning of optical sensors on a subject’s head in a lab environment [25]. In contrast, most of the remote-monitoring approaches involve the use of wireless headsets or earpieces with dry, pre-configured electrodes that can be self-applied by patients at home [26]. Commercial EEG headsets have better usability compared to traditional EEGs in terms of application time and wearing comfort [27]. The devices are also more cost-effective and more suitable for long-term monitoring compared to traditional procedures used in healthcare and research to study human sleep [28]. Additionally, remote neuroimage devices can be used to conduct neurofeedback interventions in home settings. Neurofeedback, which trains patients to volitionally regulate brain activity, is emerging as a therapeutic approach for various psychiatric and neurological conditions [29].

Integrating remote neuroimaging devices into DP presents technical and practical challenges for researchers. One fundamental challenge is the need to identify distinct patterns or features in brain activity that can serve as reliable biomarkers for specific health conditions or states of interest [30]. Moreover, while remote neuroimaging devices offer the convenience of continuous, naturalistic brain monitoring in real-world settings, they are limited in terms of the number of channels or regions they can cover compared to clinical-grade equipment used in controlled environments [31]. This restricted channel coverage can potentially overlook important brain dynamics. Furthermore, these portable devices are more susceptible to noise in the data caused by movements during recording (known as motion artifacts), which can corrupt the quality of the neural data [32]. Another challenge consists of the fact that successful implementation of remote neuroimaging for DP also depends on practical factors related to participant compliance and adherence. The ability and willingness of individuals to consistently wear and properly use these devices as intended are essential for obtaining high-quality neuroimaging data [33]. Factors such as comfort, ease of use, and perceived burden can significantly impact participant engagement and the integrity of the data collected. Researchers must carefully consider and address these human factors to ensure the feasibility and effectiveness of remote neuroimaging in DP studies. In this scoping review, we intend to comprehensively evaluate biomarkers of health conditions present in the data collected by wearable EEG, fNIRS, or PPG devices and the critical factors of feasibility, usability, and data quality that influence the effective implementation of remote monitoring using these devices, thereby guiding future research and application strategies in this emerging field. To the best of our knowledge, this is the first review to focus on remote, self-applied neuroimaging in healthcare applications. Several aspects of these studies are investigated to give a clear view of the current progress made towards applying remote neuroimaging in healthcare services and research. To this end, the health conditions studied, types of devices used for monitoring, and accuracy of remotely collected data compared to data collected in clinics or research labs are reported in this review.

## 2. Materials and Methods

The methodology for this scoping review followed the framework set forth by [34]. This approach consists of five steps: identification of the research question, identifying relevant studies, study selection, data charting, and lastly summarizing and reporting the results.

### 2.1. Identifying the Research Question

To determine the current uses of wearable neuroimaging devices outside of clinical settings for determining a subject’s health condition and evaluating the efficacy of interventions, a review was conducted to answer the following questions:What health conditions are studied remotely with wearable neuroimaging devices?Which wearable neuroimaging devices are prevalent in remote health studies?How does remotely collected neuroimaging data quality compare to that provided by collecting neuroimaging data using traditional/in-person clinical-setting devices?

### 2.2. Identify Relevant Studies

Literature search was conducted on 2 databases: PubMed and Scopus. PubMed has a wide coverage of literature in the fields of biomedical engineering and health sciences. Scopus includes records from multiple interdisciplinary research databases such as MEDLINE, Embase, and IEEE. The keywords used to research relevant studies in the mentioned databases are shown in Table 1.

### 2.3. Selection of Eligible Studies

The process of selecting eligible studies was guided by Preferred Reporting Items for Systematic reviews and Meta-Analyses (PRISMA) guidelines [35]. The titles and abstracts of the collected studies were first screened; then, a review of the full manuscripts was conducted. The included studies met all the following inclusion criteria:Related to a healthcare application;Full text was available;Concerned non-invasive neuroimaging using EEG, fNIRS, or PPG;Self-applied, unattended neuroimaging conducted outside clinics and research centers.

### 2.4. Data Charting

Guided by the research questions, the following details were extracted from the selected studies:The health conditions tracked using remote neuroimaging;Number of participants who participated in remote neuroimaging monitoring;Remote neuroimaging procedures and findings;The types of devices used to collect neuroimaging data remotely;Notes on the usability of the devices, including barriers and facilitators of use;Did the study report a comparison between mobile neuroimaging data and neuroimaging data collected at a clinic and/or research center? If so, how did the quality of mobile neuroimaging data compare to that of clinical neuroimaging data?

### 2.5. Collating, Summarizing, and Reporting the Results

Results collected from the selected studies were then presented in a narrative format. To answer the research questions, the results were divided into the following sections: high-level summaries of studies, health conditions monitored using remote neuroimaging, device characteristics, and quality of remote neuroimaging compared to clinical neuroimaging.

## 3. Results

The article selection process is shown in Figure 1. The search was conducted on 22 January 2024. A total of 1740 articles were collected (corresponding to 933 articles from Scopus and 807 articles from PubMed). After the screening and full-manuscript-review steps, 23 eligible articles were identified.

The studies were then broken down by year published and the category of the condition studied, as shown in Figure 2. The number of studies using remote EEG started increasing after 2020, which may be explained by COVID-19, which has increased attention to the necessity of remote health monitoring and telehealth [36,37].

Table 2 shows the selected articles’ titles, the condition studied, and the categories of the conditions studied. The conditions were categorized into one of the following: mental health, neurological disorders, sleep monitoring, or chronic pain.

### 3.1. Remote Monitoring Using Mobile EEG Devices

#### 3.1.1. Neurological Disorders

Recent studies have utilized EEG-based neuroimaging to identify potential biomarkers of cognitive impairment and Alzheimer’s disease. Multiple techniques have been employed, including EEG headsets, single-channel home EEGs, and ear EEG devices. Common goals were detecting abnormal patterns of brain activity associated with memory and executive function deficits.

In [45], multiple sclerosis patients were recruited to complete 10 neurofeedback training sessions over 3–4 weeks. Each training session consisted of a 3 min resting state followed by six feedback runs. During these training sessions, the participants received visual feedback of their sensorimotor rhythm power, theta power, and beta power. Increased sensorimotor rhythm power is associated with less cognitive impairment, so the purpose of the study was to train the participants to upregulate this rhythm by staying physically relaxed. Beta and theta powers were shown as controls to ensure changes in sensorimotor rhythm were not a result of motion, eye blinking, or artifacts. Half of the multiple sclerosis patients, classified as the responder group, showed significant improvements in cognitive functions, such as verbal and visual-spatial long-term memory, as well as executive functions following EEG-based neurofeedback training. Conversely, the non-responder group (half of the participants) did not demonstrate any significant cognitive changes after neuroimaging training, also lacking in their ability to modulate brain activity as intended during the sessions.

In [49], participants’ EEG data were collected at home while they slept using a single-channel EEG (employing AF7, AF8, and Fpz electrodes) for 3–6 nights. The research focused on detecting memory-relevant events from sleep EEGs, particularly the coupling of slow waves with theta bursts and sleep spindles. The results indicated that cognitive impairment was associated with lower theta burst spectral power in slow wave–theta burst coupling. Furthermore, cognitively unimpaired individuals showed less precision (regarding timing and frequency consistency) in the coupling between theta bursts and high-frequency slow waves and between sleep spindles and low-frequency slow waves. Significant correlations were found between the precision of these couplings, detected using the single-channel EEG, and Alzheimer’s biomarkers like amyloid beta and tau proteins in the cerebrospinal fluid.

In [56], Alzheimer’s disease patients underwent up to three ear-EEG recording sessions, each lasting up to 48 h and spaced three months apart over a six-month period. The recordings were taken during the daytime and while the participants slept. After removing artifacts from the EEG data, the ear-EEG device captured 27.3 h of EEG recordings per participant. The electrodes within the ear canal (ELE/ELI and ERE/ERI) demonstrated the best data quality, likely due to lower impedance in the moister environment of the ear.

#### 3.1.2. Mental Health

In [40], the Muse 2 device was used to measure daily changes in individual alpha frequencies (IAF). The participants of this study were asked to undergo two high-density EEG (HD-EEG) sessions and fill in self-report questionnaires at a lab. Between the in-person sessions, they were also asked to use the Muse 2 device at home daily for 4 weeks. The EEG recording sessions conducted at home and lab consisted of 3 min eyes-open and 2 min eyes-closed resting state recordings. Statistically significant correlations were found between the variability of IAF as recorded by TP9 and TP10 electrodes of the Muse 2 devices and the State–Trait Anxiety Inventory Trait Anxiety Score. This suggests that the variability of IAF itself, rather than just its average value, could be an important marker related to anxiety. Moreover, no significant relationships were found between the number of low-quality recording sessions and the anxiety scores.

#### 3.1.3. Sleep Monitoring and Disorders

Various studies have utilized remote/at-home EEG monitoring devices to assess sleep patterns. In these studies, training protocols were put in place to improve the quality of data and lower the failure rates of at-home EEG sleep data captured by the participants. Remote EEG monitoring was able to monitor sleep stages (rapid eye movement (REM), non-REM stage N1, non-REM stage N2, and non-REM stage N3) with an accuracy similar to that of lab polysomnography (PSG).

The results of a case study are reported in [38], wherein a subject was asked to wear an ear-EEG device during (1) 40 work shifts at night and (2) for five nights when the subjects slept at home. During the nights that did not involve work shifts, PSG was also conducted to serve as a baseline for the EEG data. The data recorded from the ear-EEG were then successfully used to classify the sleep stages: REM, non-REM stage N1, non-REM stage N2, and non-REM stage N3. The data captured using the ear-EEG device were also used to classify shift nights, nights when the patients slept at home, and sleep after a shift night with 87% accuracy. This highlights the ability of wearable EEGs to monitor sleep with a quality similar to that of PSG while reducing study costs and improving the patient’s comfort.

In [39], owners of the Dreem Headband who did not suffer from sleep issues were recruited. In total, 470 nights of data captured using Dream Headbands were used in an analysis conducted using the processed sleep stages recorded at each timestamp. This study found significant night-to-night variability in sleep parameters among participants on different nights (regarding total sleep time; sleep onset latency; waking after sleep onset; percentage of time spent in sleep stages N1, N2, N3, and REM; latency of the onset of the different sleep stages; latency of persistent sleep; awakenings; and sleep efficiency), highlighting the need to obtain multiple-night recordings to accurately assess sleep patterns in clinical trials.

In [43], the participants were asked to participate in one night in-lab PSG and two at-home EEG-monitoring sessions using the Dreem 2 Headband. Simulatnesously, the participants were asked to put on other wearable devices (rings or wristbands). The purpose of this study was to evaluate the performance of a personalized algorithm trained on data from rings and wristbands in detecting sleep stages. The Dreem 2 Headband data were used as a gold standard for at-home sleep monitoring, highlighting an application of remote neuroimaging devices in research. Moreover, it was reported that at-home EEG sleep recordings revealed higher percentages of REM and deep-sleep stages compared to in-lab recordings, suggesting the superior effectiveness of naturalistic at-home EEG monitoring compared to lab PSG.

In [44], the participants conducted a minimum of one night of PSG at home using the Nox A1 recorder. Training and detailed instructions were provided to ensure accurate data capturing. The participants who were not successful in obtaining a PSG recording on their first try were asked to participate in a second night of at-home sleep monitoring. This study demonstrated the potential to link sleep abnormalities with health outcomes among individuals with and without human immunodeficiency virus (HIV).

In [46], participants used Neuroon (an EEG mask design for sleep monitoring), Fitbit Charge, and a medical-grade single-channel EEG device called Sleep Scope for three consecutive nights. Comparing the performance of Neuroon with that of Sleep Scope indicated that Neuroon’s effectiveness in tracking sleep parameters such as total sleep time, sleep efficiency, and waking after sleep onset varies with signal quality. With good signal quality, Neuroon showed comparable results to the medical device for some parameters, but discrepancies increased in cases of poor signal quality. This study also highlights the advantages of using wearable EEG devices over wearable wristbands in monitoring sleep with respect to aspects such as total awake time and sleep onset latency.

In [47], the participants were trained to use at-home sleep-monitoring devices, including a single-channel EEG and a home sleep test (HST) device. They were trained using educational manuals and a 2 h training session on device usage. The implementation of the protocol improved the quality of sleep data, reducing the failure rate from approximately 40 to 50% to 19%.

In [50], participants who were undergoing gender-affirming hormone therapy (GAHT) were asked to record their sleep at home using a Smartsleep device. Smartsleep has one EEG electrode (M1). Remote EEG monitoring was performed at baseline and after 3 months of hormone therapy. The key outcomes measured using a remote EEG included sleep architecture changes such as sleep onset latency, total sleep time, waking after sleep onset, sleep efficiency, number of interruptions, number of arousals, slow-wave sleep duration and percentage, REM sleep latency, and REM sleep duration. For transmasculine participants, significant decreases in slow-wave sleep duration and percentage as well as a significant decrease in REM sleep latency and an increase in REM sleep duration were observed. No statistically significant changes were observed in the transfeminine participants’ sleep architecture after 3 months of GAHT when compared to the baseline measurements.

In [51], the participants conducted 20 nights of remote monitoring using both PSG and an ear-EEG and then 100 nights of monitoring exclusively with an ear-EEG. Prior to data collection, each participant underwent a fitting session for the earpieces used in the study. The ear-EEG device had four electrodes. The data from both PSG and the ear-EEG were used to detect sleep stages effectively, showing that the ear-EEG did not result in a statistically significant effect on sleep stage detection.

In [52], two groups of participants (n = 12, each) conducted either lab-based PSG along with monitoring using a trEEGrid device (pre-applied) or self-applied monitoring using a trEEGrid device at home. trEEGrid consists of nine single-use self-adhesive gel electrodes; seven of the nine electrodes provide EEG recordings. The data from both devices were processed to detect the sleep stages, reporting a Cohen’s Kappa equal to 0.70 between lab PSG and trEEGrid.

In [54], the participants conducted three nights of sleep monitoring using one of two devices that had two-channel EEG electrodes either fixed on the forehead or behind the ear on the mastoid. This study revealed distinct “Hi Deep” and “Lo Deep” sleep stages associated with different brain frequencies and electrodermal activity. “Lo Deep” sleep was particularly linked to increased electrodermal activity, suggesting it may have unique physiological significance.

In [55], the participants used novel flat-type dry electrodes and traditional pre-gelled electrodes to obtain overnight sleep EEG recordings simultaneously for one night. The data from both sensors were processed and used to classify sleep stages. The analysis of sleep microstructure, such as slow-wave activity and spindles, showed that both dry and pre-gelled electrode types could effectively monitor sleep stages and microstructures, with a kappa value equal to 0.66. Dry electrodes were subject to more artifacts caused by sweat.

In [58], maritime pilots used Smartsleep for 14 consecutive nights (7 of which were workdays, and the rest were days off) to obtain EEG recordings during sleep. Sleep quality was assessed subjectively using the PSQI. The findings revealed a trend towards increased efficiency in provoking deep sleep during work weeks, with a 0.6% increase in deep sleep time per day, although this was not statistically significant (*p* = 0.08).

In [59], 63 participants were instructed to self-apply an EEG headset called Sleep Profiler at home for two consecutive nights. Sleep Profiler acquires EEG readings from three channels: AF7-AF8, AF7-Fpz, and AF8-Fpz. The EEG data were then processed to calculate sleep stages and sleep biomarkers for each night (such as sleep time, latency, cortical arousal index, and spindle durations). Certain sleep biomarkers were associated with self-reported medications, showing enhanced spindle activity and elevated sigma and beta power in sleep stages N2 and N3 among antidepressant users (*p* < 0.01). In contrast, individuals on antihypertensive medications experienced a decrease in deep sleep (N3), an increase in lighter sleep (N2), and lower levels of delta, theta, alpha, and sigma power during these stages (*p* < 0.001).

### 3.2. Remote Neurofeedback Interventions

At-home, EEG-based neurofeedback interventions have been proven to improve symptoms for a range of conditions, including attention deficit hyperactivity disorder (ADHD), post-traumatic stress disorder (PTSD), sleep disorders, and chronic pain. The key advantages of home-based approaches highlighted here include increased access to therapy, similarity in outcomes, and reduced interference in patients’ lives compared to facility-based application.

In [41], the authors reported the results of an 8-week brain–computer-interface-based (BCI) intervention for ADHD in children between 6 and 12 years old. The BCI consists of a dry electrode EEG, which measures participants’ attention and adjusts their score in a game based on their ability to pay attention to the game during the training session. During these 8 weeks, the participants were split into two groups (lab and home), and each group was asked to complete 24 training sessions. Both groups were evaluated based on the ADHD Rating Scale (ADHD-RS) by parents and clinicians. Both groups showed improved scores and no statistically significant differences between them. This suggests the home-based ADHD intervention was as effective as the lab-based one.

In [42], a crossover trial with 68 adults evaluated the effects of auditory EEG neurofeedback from the Muse headband on state mindfulness and meditation experiences during focused-attention meditation. Receiving auditory feedback was associated with a near-significant 15% increase in correct breath counting, suggesting improved state mindfulness. Feedback also led to significantly lower device-measured mind wandering but had negligible effects on recovery from mind-wandering episodes. Qualitative analysis revealed that using feedback obtained via Muse at home was seen as both helpful for guiding meditation and unhelpful due to being stressful, distracting, or incongruent with subjective experience. Moreover, this study reported a high adherence rate of 81% with respect to using Muse at home over the period of 14 days.

In [53], a 4-month neurofeedback intervention was conducted for veterans with a traumatic brain injury (TBI) or PTSD. The intervention consisted of four weekly sessions conducted at home during which the participants used an app that provides auditory feedback in the form of relaxing sounds associated with brainwave patterns of relaxed brains. This training aimed to reinforce alpha brain waves that are associated with reduced pain. Sixty-seven percent of all the training sessions (n = 965) were reported to reduce pain after 10 min of mobile neurofeedback. Moreover, statistically significant reductions in PSTD symptoms, Patient Health Questionnaire score (PHQ-9) pain intensity, pain interference, depressive symptoms, suicidal ideation, anger, and sleep disturbances were reported between the baseline and follow-up (post-neurofeedback intervention).

In [48], the participants received sensorimotor rhythm (SMR) neurofeedback using the URGOnight device to test its efficacy in treating sleep problems. The URGOnight device has two EEG channels (C3 and C4). The participants either engaged in 40 (n = 21) or 60 (n = 16) training sessions and filled out the Holland Sleep Disorders Questionnaire (HSDQ) according to the Pittsburgh Sleep Quality Index (PSQI). The training consisted of resting-state recording with eyes closed (30 s), resting-state recording with eyes open and then closed (30 s), and five 30 min feedback runs when the participants received visual feedback of their SMR power. This intervention led to a notable improvement in sleep quality among the participants, with a statistically significant reduction in PSQI scores, indicating better sleep health. Moreover, there was an increase in sleep duration, averaging a half-hour gain after completing the neurofeedback intervention.

In [57], a Dreem Headband was used to evaluate the effectiveness of a slow-oscillation (SO) stimulation intervention to improve N3 sleep stage quality. The simulation was delivered via the Dreem Headband once N3 sleep stage was detected. The intervention aimed to increase delta power sleep stage, since delta power is associated with higher sleep quality during N3. The results showed a statistically significant increase in delta band power after simulation delivered by the headband compared with that for sham triggers.

In [60], participants with chronic pain were asked to complete 4–6 neurofeedback training sessions per week (for 8 weeks) using an Axon EEG headband and an associated tablet application. The training sessions started with a 2 min resting state recording (eyes open) followed by another 2 min resting state recording (eyes closed) that were used to establish a baseline for relative alpha, theta, and hi-beta thresholds. Next, each participant chose one of three games that would progress based on the real-time alpha power. The aim of this neurofeedback was to teach the participants to upregulate their alpha bands, which are associated with relaxation. Sixty-nine percent of the participants reported an improvement in pain after the completion of the neurofeedback intervention using an Axon EEG headband. Statistically signification improvements in central sensation, sleep quality, anxiety, and depression were also found as secondary measures of the study post intervention. The remote EEG records showed a statically significant increase in relative alpha bands (associated with relaxation) and a decrease in hi-beta bands (associated with anxiety). Additionally, 75% of the participants reported improvement in quality-of-life post intervention determined using the EuroQol 5-Dimension 5-Level (EQ-5D-5L) questionnaire, which measure health-related quality of life.

### 3.3. Mobile Device Charastristicts

All the studies in this review used a form of mobile EEG device. Table 3 shows the types of remote neuroimaging devices and the number of studies in which they were used.

The Muse 2 device (an EEG headband) was used in [40]. NeuroSky Mindwave (an EEG headband) was used in [53]. The Axon EEG neurofeedback system (an EEG headband) was used in [60]. A BCI (an EEG headband) was used in [41]. A Dreem headband (EEG-headband) was used in [39,57]. The Dreem 2 headband (EEG-headband) was used in [43]. The URGOnight device (an EEG headband) was used in [48]. Smartsleep (an EEG headband) was used in [50,58].

Regarding the usability and feasibility of monitoring EEGs remotely using these headbands, ref. [40] reported that 72% of participants had 75% or more of their at-home recording sessions included in the final analysis. The participants were shown how to wear the Muse 2 device during the first in-person session. The majority of the participants thought using Muse 2 was straightforward (89%) and did not interfere with their daily life. Eighty percent of the parents in [41] said their children could set up the EEG headband with little or no supervision at home. While only 60% of the home-based group had a uniform spread of training sessions in 3 or more weeks (which can be compared to 90% in the lab group), both groups achieved similar ADHD-RS improvements. The authors of [53] reported that 36 participants were able to record a mean of 33.09 (SD = 30.73) 10 min training sessions over the course of 4 months using the NeuroSky Mindwave EEG headband. The participants of this study reported they preferred using the headset compared to medications. The authors of [60] reported that participants were able to use Axon EEG neurofeedback headbands on their own after a remote session in which they were taught how to set up the headset and use the associated tablet application. No further usability metrics were reported in this study. In [39], the participants were already owners of the Dreem Headband before the beginning of the study.

A research-grade biofeedback device, Mind Media NeXus-10, was utilized in [45]. The participants of this study received instructions on how to set up the system before the beginning of the neurofeedback training sessions.

A single-channel EEG device worn on the forehead was used in [49] to track participants during their sleep. This study did not provide information on whether the participants were trained with respect to how to use the device or other usability metrics. A single-channel EEG was also used in [47], achieving 81% success rates in recording home sleep tests. Two-channel EEG devices (worn on the forehead or behind the ear) were used in [54]. The participants were given instructions on setting up the devices and electrodes themselves. They reported falling asleep within 25 min while having the devices on.

An ear-EEG device with dry electrodes was utilized in [38] during the subjects’ sleep and night work shifts. This device was self-applied and did not require maintenance except for charging the battery. The subjects were able to record 35 nights on their own, while 13 nights were not recorded due to various reasons (discomfort, technical difficulties, and forgetting to charge the battery). An ear-EEG was also used in [51] to monitor participants while sleeping. The participants assessed the comfort and usability of the ear-EEG device through a questionnaire completed each morning. After nights where a PSG device was used alongside an ear-EEG, most of the sleep quality ratings were “bad” or “bearable”. Conversely, when they only used the ear-EEG device, 62% of the responses rated sleep quality as “good” or “very good”. The authors of [56] conducted a feasibility study for using an ear-EEG device to monitor the EEG data of Alzheimer’s disease patients. Half of the participants reported difficulties with device placement and wearability. Four participants noted the EEG recorder’s weight and comfort were problematic when worn around the neck during daily activities. Seven participants experienced mild ear tenderness.

A PSG recorder, Nox A1, was used in [44]. Out of 960 participants, 851 (88.6%) successfully completed the sleep study on their first or second attempt. Among the successful recordings, 72.4% were rated as excellent quality, and 13.9% were very good. This indicates that the procedures for self-applied PSG were easily adoptable by the participants. In [52], a self-application study revealed high usability, with most participants finding the process of applying the EEG system easy or very easy with respect to various aspects, such as putting on the chest belt, preparing the skin, finding the correct electrode positions, and connecting the electrode grid to the amplifier. However, some participants faced difficulties, particularly with positioning electrodes behind the ear. The setup time for the self-application varied among participants, with a mean time of 12 min, and impedance levels measured after 20 min of wear showed no significant difference compared to the lab-applied setups, demonstrating that the participants could successfully self-apply the system with impedance levels suitable for reliable data collection.

### 3.4. Quality of Mobile EEGs Compared to Clinical EEGs

Table 4 shows data quality evolution results reported in the selected studies. Statistical tests were used to evaluate the quality of data collected remotely by wearable EEG devices by comparing them to EEG data collected in clinics or using traditional EEG devices.

In [40], a thorough comparison of in-lab HD-EEGs and at-home EEGs was conducted to evaluate the data quality of the latter. First, the data from electrodes AF7 and AF8 conducted during at-home recordings were excluded from the analysis due to high variability across devices and recording sessions. On the other hand, no statistically significant difference was found between IAF according to electrodes TP9 and TP10 during at-home recordings and data from HD-EEG across different recording sessions. Thus, there were no biases in the data recorded at home using Muse 2 in this study. One important consideration in this study is that gel was applied to the Muse 2 electrodes, and this application was reported to improve the quality of data recording.

In [38], the quality of ear-EEG data was evaluated using PSG data as a baseline. The PSG data consisted of 10 EEG scalp electrodes (F3, F4, C3, C4, O1, O2, M1, M2, and Fpz) as well as electrooculogram and electrocardiogram (ECG) electrodes and two respiratory belts. After processing the EEG data and using them to determine the sleep stages from both ear-EEG and PSG information, a 0.72 Cohen’s kappa was achieved for six different categories (awake, REM, non-REM stage N1, non-REM stage N2, non-REM stage N3, and unknown). The authors reported this agreement between the algorithm-scored ear-EEG data and the manually scored PSG data was substantially high.

In [51], data quality was primarily assessed through the artifact rejection process, which, on average, excluded 10.2% of the data before channel selection and 4.4% afterward. The artifacts were categorized into device-related noise (3.2%), spikes (2.6%), high-frequency (3.9%), and high-amplitude (0.5%). The sleep stages detected using both ear-an EEG and PSG were not significantly different from each other.

In [52], analysis of hypnograms from both the trEEGrid device and commercial PSG systems showed varying degrees of agreement among the participants, with Cohen’s kappa values ranging from 0.58 (moderate agreement) to 0.83 (almost-perfect agreement). Overall, this study found substantial agreement in sleep stage annotations between the systems, with a Cohen’s kappa of 0.70 ± 0.01. Impedance levels significantly dropped by the end of the night, indicating that the quality of the data remained reliable despite initial disconnections.

In [59], the first stage of the study consisted of simultaneous in-lab PSG and EEG recording using a Sleep Profiler (n = 47). The PSG data were reviewed by five PSG technicians independently to label the sleep stages, while the EEG recordings were processed automatically. The sleep stages chosen by the majority of the PSG technicians were then compared to the automatically chosen sleep stages from the EEG data. The results showed that the sleep stages drawn from the EEG data had an accuracy higher than 80% across different sleep stages except for the N1 sleep stage (32% accuracy). Different autoscoring methods also had less significant differences from PSG technicians in the percentage of time N1 and N3 stages were selected.

The following studies did not report a comparison of data quality between model and clinical EEGs: [39,41,42,43,44,45,46,47,48,49,50,53,54,57,58,60].

## 4. Discussion

This scoping review aimed to summarize the current applications of wearable neuroimaging devices, particularly EEG headsets, in remotely monitoring health conditions outside of clinical settings. Prior research has established the value of EMA for psychiatric and neurological disorders, and this value is enabled by wearable sensors [2]. Mobile EEG devices have been leveraged to monitor a variety of health issues remotely, including mental health disorders, neurological disorders, sleep abnormalities, and chronic pain management. The applications of these devices span conditions such as ADHD, anxiety, Alzheimer’s disease, and multiple sclerosis, and they can also be used for sleep monitoring. EEG headsets have also been used to conduct interventions using neurofeedback. Our review found that mobile EEG sensors were also used to deliver neurofeedback interventions in home settings for disorders including ADHD and PTSD, pain, improving mindfulness, and improving sleep quality. The feasibility of remotely monitored neurofeedback indicates potential to enhance access to brain-based therapies.

The reviewed applications of remote EEG tracking in health monitoring focused on measuring the EEG frequency bands such as alpha, beta, theta, and delta bands to assess health conditions. Moreover, proprietary EEG processing algorithms were utilized in some studies to detect sleep stages and shown to yield similar accuracy to lab PSG. Similarly, remote neurofeedback interventions provided real-time feedback during the training by tracking changes in specific frequency bands associated with the targeted conditions [61]. Separating the EEG signal into different frequency bands is referred to as band analysis. Other approaches to analyzing EEG data include event-related potentials (analyzing the brain’s response to specific stimuli or events) and connectivity analysis (examining the functional and effective connectivity between different brain regions) [14]. An opportunity exists to leverage these approaches for application to data from mobile EEG devices, allowing remote monitoring of a wide verity of neurological conditions, cognitive processes, and brain networks.

Another opportunity exists with respect to expanding remote monitoring to fNIRS and PPG, for which data are absent in the current literature. fNIRS allows researchers to study how activity in specific brain regions is related to various health conditions and behaviors [62]. Moreover, combining fNIRS with an EEG provides a multimodal approach that capitalizes on the strengths of each technique [63]. An EEG can detect changes in brain electrical activity very quickly (high temporal resolution), while fNIRS provides better spatial information about which specific brain regions are active (high spatial resolution). By combining these two techniques, researchers can track both the timing of brain activity and the location of this activity at the same time with good precision. This gives a more complete picture of how the brain’s electrical signals and blood flow work together in thinking, behavior, and brain disorders [64]. PPG is also recognized as being able to complement EEG data in monitoring different health conditions such as those related to mental health [65]. Some wearable EEG headbands, such as Muse, and other wearable devices already incorporate PPG sensors, constituting a big facilitator for tracking and analyzing data.

Consumer-grade EEG headsets are the most widely used type of wearable neuroimaging device, present in over half of the reviewed studies. EEG headsets, however, suffer from signal quality issues, highlighted especially during sleep monitoring. Thus, validation of remotely collected neuroimaging data quality against clinical standards remains crucial, particularly as machine learning techniques are applied for automated health monitoring and diagnostics [66]. Ear-EEG devices are more discreet compared to EEG headsets and yet offer comparable data quality during sleep and day-to-day activities. Additionally, the moist environment inside the ear can improve the performance of dry EEG electrodes by enhancing conductivity and signal quality. However, participants who used an ear-EEG sometimes reported discomfort associated with wearing the ear-EEG devices for a long time. Moreover, an ear-EEG offers restricted coverage of brain regions, so the EEG monitoring scheme needs to be compatible with ear-EEG coverage [67].

Neurofeedback interventions were proven in the results to be promising therapies that can be delivered remotely in the home setting. Delivering neurofeedback remotely enhances access to these brain-based interventions outside of clinics and allows ecological training in natural environments, leading to less interference with patients’ lives and increasing their ability to transfer what they learned to other tasks [68]. Moreover, neurofeedback is user-friendly and can effectively guide users by providing feedback when EEG signals are suboptimal, ensuring meaningful engagement and accurate data collection for therapeutic purposes [69].

While fewer than half of the studies directly compared data quality between mobile and clinical EEG acquisition, those that did found moderate to substantial agreement overall in sleep staging and neural signal detection. Mobile EEGs were successfully validated against HD-EEGs, PSG, and research-grade systems, with some caveats around reduced channel coverage and signal artifacts. This indicates remote EEGs can approximate in-lab protocols for capturing relevant brain activity patterns. However, rigorous quality testing is still needed, especially for clinical diagnostic purposes.

Ensuring reliable signal quality emerges as a critical challenge for remote neuroimaging with wearable EEG devices. Consumer-grade headsets often suffer from limited channel coverage, increased susceptibility to motion artifacts, and vulnerability to environmental noise interference, compromising data fidelity compared to clinical-grade systems. Rigorous signal-processing techniques, such as advanced artifact removal algorithms and adaptive filtering methods, are imperative to mitigate these issues and extract accurate neural signals in real-world settings [70]. Moreover, robust validation strategies are necessary to establish statistically significant agreement between remotely acquired data and clinical neuroimaging standards, particularly for diagnostic applications [71,72]. Overcoming these signal quality challenges through innovative hardware designs, optimized electrode configurations, and sophisticated computational methods remains a key technical hurdle to unlocking the full potential of wearable EEG devices for continuous, ecological brain monitoring outside traditional laboratory environments.

## 5. Conclusions

The motivation behind this scoping review was to investigate the emerging role of wearable neuroimaging technologies for remote patient monitoring outside of clinics. Studies that leveraged mobile brain-scanning devices to assess health conditions in real-world settings were included. The key research questions focused on which conditions were evaluated remotely, which types of mobile devices were utilized, and how the data quality compared to clinical standards. The findings revealed a breadth of applications across mental health, neurological, sleep, and pain disorders using primarily consumer-grade EEG headsets that provided reasonable signal fidelity. However, there is still work remaining in terms of improving device usability, validating their performance against lab-based protocols, and ensuring ethical data practices. Overall, this review highlights the promising capacity of portable brain-imaging devices to monitor patients continuously in natural environments while underscoring important considerations as this technology evolves.

## Figures and Tables

**Figure 1 biomimetics-09-00237-f001:**
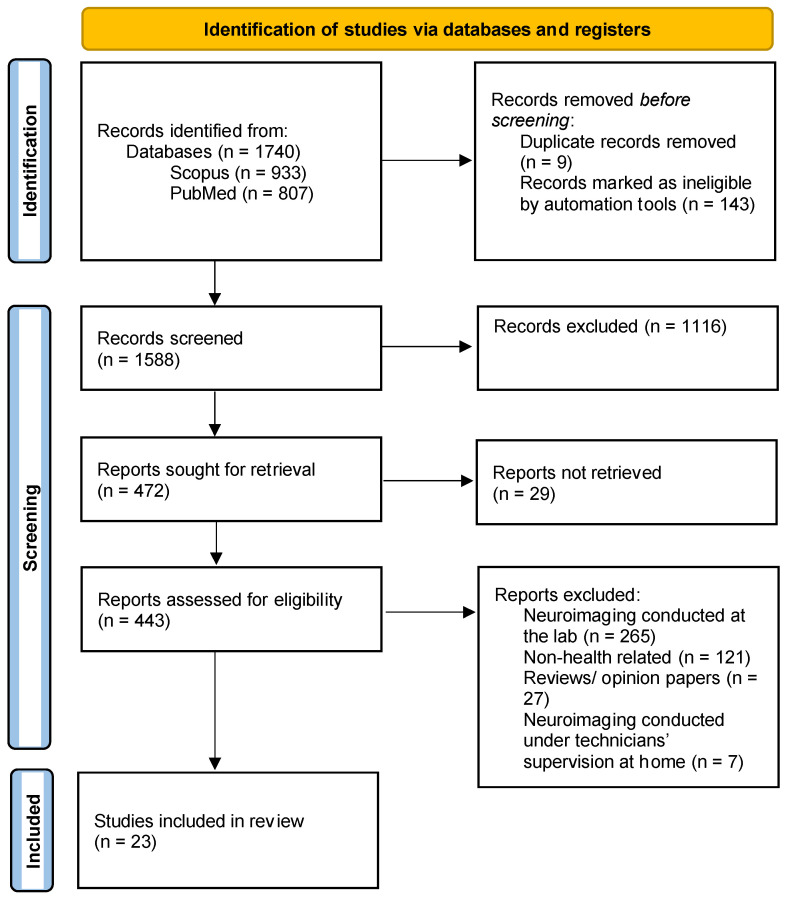
Article selection process flowchart modified from suggested template in PRISMA.

**Figure 2 biomimetics-09-00237-f002:**
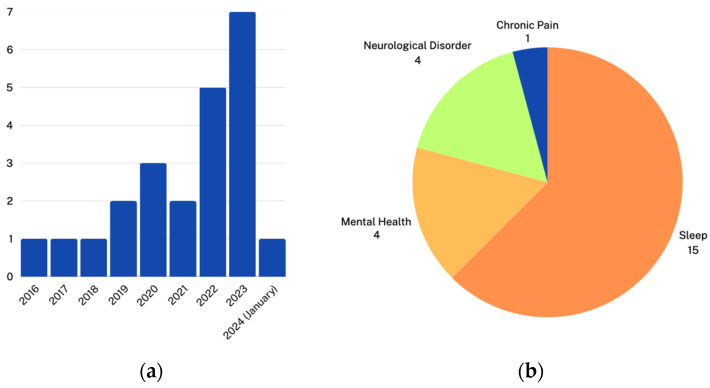
Breakdown of the eligible papers by (**a**) year published and (**b**) category of condition studied.

**Table 1 biomimetics-09-00237-t001:** Databases and keywords used in literature search.

Database	Keywords
PubMed/Scopus	(“Neuroimaging Devices” OR EEG OR fNIRS OR PPG OR “brain recording” OR “brain monitoring” OR neurosensory) AND (wearable OR continuous OR remote OR wireless OR headband OR “home-based” OR “Home environment” OR “home monitoring” OR “Consumer-Grade” OR Portable OR Mobile) AND (“Healthcare Applications” OR “Mental Health” OR “Neurological Disorders” OR “Psychiatric Diseases” OR “Sleep Disorders”)

**Table 2 biomimetics-09-00237-t002:** Selected studies’ details with their respective references.

Article Title	Condition or Physiological Measurement	Category of the Condition	Number of Participants *	Reference
Long-term ear-EEG monitoring of sleep—A case study during shift work	Sleep stage classification	Sleep motoring	1	[38]
Objective multi-night sleep monitoring at home: variability of sleep parameters between nights and implications for the reliability of sleep assessment in clinical trials	Sleep stage variability	Sleep motoring	94	[39]
Day-to-day individual alpha frequency variability measured by a mobile EEG device relates to anxiety	Anxiety	Mental health	18	[40]
Home-based brain–computer interface attention training program for attention deficit hyperactivity disorder: a feasibility trial	ADHD	Mental health	10	[41]
EEG neurofeedback during focused attention meditation: Effects on state mindfulness and Meditation Experiences	Mindfulness	Mental health	29	[42]
Performance of a multisensor smart ring to evaluate sleep: in-lab and home-based evaluation of generalized and personalized algorithms	Sleep stage classification	Sleep motoring	36	[43]
Methods for home-based self-Applied polysomnography: The Multicenter AIDS Cohort Study	Sleep abnormalities in HIV patients	Sleep motoring	960	[44]
Self-regulation of brain activity and its effect on cognitive function in patients with multiple sclerosis—First insights from an interventional study using neurofeedback	Multiple Sclerosis	Neurological disorder	14	[45]
Validity of Consumer Activity Wristbands and Wearable EEG for Measuring Overall Sleep Parameters and Sleep Structure in Free-Living Conditions	Sleep stage classification	Sleep motoring	25	[46]
Multi-modal home sleep monitoring in older adults	Sleep motoring	Sleep motoring	29	[47]
Evaluation of the URGOnight Tele-neurofeedback Device: An Open-label Feasibility Study with Follow-up.	Sleep quality	Sleep disorders	37	[48]
Mapping sleep’s oscillatory events as a biomarker of Alzheimer’s disease.	Alzheimer’s Disease	Neurological disorder	205	[49]
Influence of sex hormone use on sleep architecture in a transgender cohort.	Sleep quality	Sleep disorders	73	[50]
At-home sleep monitoring using generic ear-EEG	Sleep stage classification	Sleep motoring	10	[51]
Pre-gelled Electrode Grid for Self-Applied EEG Sleep Monitoring at Home	Sleep stage classification	Sleep motoring	12	[52]
Mobile Neurofeedback for Pain Management in Veterans with TBI and PTSD	Depression, anger, sleep disturbance, suicidal ideation, and chronic pain	Mental health	36	[53]
Visualization of whole-night sleep EEG from 2-channel mobile recording device reveals distinct deep sleep stages with differential electrodermal activity	Sleep stage classification	Sleep motoring	51	[54]
A Protocol for Comparing Dry and Wet EEG Electrodes During Sleep.	Sleep stage classification	Sleep motoring	4	[55]
Long-Term EEG Monitoring in Patients with Alzheimer’s Disease Using Ear-EEG: A Feasibility Study	Alzheimer’s Disease	Neurological disorder	10	[56]
Performance of an Ambulatory Dry-EEG Device for Auditory Closed-Loop Stimulation of Sleep Slow Oscillations in the Home Environment	Sleep quality	Sleep disorders	90	[57]
Home-EEG assessment of possible compensatory mechanisms for sleep disruption in highly irregular shift workers—The ANCHOR study	Sleep quality	Sleep disorders	10	[58]
The Accuracy, Night-to-Night Variability, and Stability of Frontopolar Sleep Electroencephalography Biomarkers	Sleep monitoring	Sleep motoring	63	[59]
Home-Based EEG Neurofeedback Intervention for the Management of Chronic Pain	Chronic Pain, depression, anxiety, and quality of life score	Pain management	16	[60]

* Number of participants who were subjected to a remote EEG.

**Table 3 biomimetics-09-00237-t003:** The types of remote neuroimaging devices and the number of studies in which they were used.

Device Type	Number of Studies
EEG headset	12
Ear-EEG	3
Medical-grade device (EEG)	3
Single-channel EEG	2
EEG mask	1
2-channel EEG	1

**Table 4 biomimetics-09-00237-t004:** Data quality assessment of remotely collected EEG data compared to EEG data collected using traditional methods.

Data Quality Test	Data Quality Assessment	Gold-Standard Data	Reference
Bayesian *t*-test on spectral correlations	High correlation for TP9 and TP10 channels. Low correlation for AF7 and AF8 channels.	HD-EEG data collected at the clinic	[40]
Agreement in sleep stage classification	High (Cohen’s Kappa = 0.72)	Scalpe EEG collected at home	[38]
Linear mixed model on sleep stage classification	No statically significant difference was found between types of equipment.	PSG data collected at home	[51]
Agreement in sleep stage classification	Moderate to high (Cohen’s Kappa = 0.58 to 0.83)	PSG data collected at the clinic	[52]
Agreement in sleep stage classification	High agreement between expert-reviewed PSG and automatically detected sleep stages from EEG data (higher than 80% for all sleep stages except N1, with a mean Cohen’s Kappa = 0.67)	PSG data collected at the clinic	[59]

## Data Availability

No new data were created or analyzed in this study. Data sharing is not applicable to this article.

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
