# Peer review of "Remote Wearable Neuroimaging Devices for Health Monitoring and Neurophenotyping: A Scoping Review"

_biomimetics, 2024, doi:10.3390/biomimetics9040237_

Round 1
Reviewer 1 Report
Comments and Suggestions for Authors
Emish et al. Remote Wearable Neuroimaging Devices for Health Monitoring and Neurocognitive Phenotyping: A Review Scoping Review
This review emphasize on EEG, fNIRS or PPG devices, to monitor brain’s activity remotely, guiding future research and application strategies in this emerging field. The article is confusing and requires refinement. My recommendations are outlined below: The article lacks comprehensive insights into all technologies presently utilized in the wearable device’s ecosystem. Although the abstract emphasizes EEG, fNIRS, and PPG devices, literature is only found on EEG-based devices, which needa thorough discussion. Technical challenges could be addressed briefly. I believe the article could significantly benefit from the inclusion of more sources and in-depth discussions on the findings, rather than merely listing references.
Author Response
For review article
|
Response to Reviewer 1 Comments
|
|
Comments 1: This review emphasize on EEG, fNIRS or PPG devices, to monitor brain’s activity remotely, guiding future research and application strategies in this emerging field. The article is confusing and requires refinement.
|
|
Response 1: Thank you for pointing this out. We agree with this comment. Therefore, we edited the introduction to provide a clearer context for this work. We've enriched the introduction with a definition of Digital Phenotyping (DP) to establish a foundation for understanding its role in monitoring health and behavior through digital tools.
“Digital Phenotyping (DP) is a field that employs digital tools to collect data on individuals' health and behavioral patterns, providing insights into human well-being [1]. At the heart of DP is passive data collection, where sensors and software in personal devices such as smartphones and wearables automatically record various health-related metrics without intervening in the users’ lives [2]. This approach allows for the continuous and real-time monitoring of health indicators, thereby generating a comprehensive view of an individual's health status [3]. The data obtained through passive collection is instrumental in personalized public health, where it is used to design health interventions and programs tailored to individual profiles [4]. DP plays a critical role in the development of personalized, data-driven health strategies, aiming to improve preventive care and health promotion at both individual and community levels [5]. Emerging technologies in DP are enabling the monitoring of brain activity, providing valuable insights into cognitive functions, emotional states, and neurological conditions [6,7]. By integrating brain activity data with other digital biomarkers, researchers aim to develop comprehensive neurological profiles that can inform personalized interventions and treatments for conditions such as depression, anxiety, and neurodegenerative disorders. Ecological momentary assessment (EMA), a research methodology that involves observing subjects in their natural environment [8], addresses the limitations of traditional neuroimaging techniques, which often lack real-world applicability [9].” (page 1, paragraph 2)
Moreover, we elaborated on the technical challenges of incorporating remote neuroimaging into digital phenotyping. These challenges shaped the research questions that our review aims to answer.
“Integrating remote neuroimaging devices into DP presents technical and practical challenges for researchers to overcome. One fundamental challenge is the need to identify distinct patterns or features in brain activity that can serve as reliable biomarkers for specific health conditions or states of interest [30]. Moreover, while remote neuroimaging devices offer the convenience of continuous, naturalistic brain monitoring in real-world settings, they are limited in terms of the number of channels or regions they can cover compared to clinical-grade equipment used in controlled environments [31]. This restricted channel coverage could potentially overlook important brain dynamics. Furthermore, these portable devices are more susceptible to noise in the data caused by movements during the recording (known as motion artifacts), which can corrupt the quality of the neural data [32]. Another challenge is that successful implementation of remote neuroimaging for DP also depends on practical factors related to participant compliance and adherence. The ability and willingness of individuals to consistently wear and properly use these devices as intended is essential for obtaining high-quality neuroimaging data [33]. Factors such as comfort, ease of use, and perceived burden can significantly impact participant engagement and the integrity of the data collected. Researchers must carefully consider and address these human factors to ensure the feasibility and effectiveness of remote neuroimaging in DP studies.” (page 2, paragraph 4)
Comments 2: The article lacks comprehensive insights into all technologies presently utilized in the wearable device’s ecosystem.
Response 2: We thank you for your comment. We have integrated background information on EEG, fNIRS, and PPG as non-invasive neuroimaging modalities.
“Electroencephalogram (EEG), functional Near Infra-Red Spectroscopy (fNIRS), and Photoplethysmography (PPG) are non-invasive neuroimaging modalities well-suited for continuous brain monitoring outside traditional lab settings, due to their portability, cost effectiveness and minimal interference on the patient’s life [13]. EEG measures the electrical activity of the brain by recording the voltage fluctuations resulting from ionic current flows within the neurons of the brain [14]. It captures the electrical activity of the brain at various frequency bands, where different frequencies act as biomarkers of different health conditions. For example, alpha waves, which are in the relatively high frequency range (8-12 Hz), are associated with relaxation and calmness. In contrast, beta waves, which are lower in the frequency (range of 12-30 Hz) are associated with high mental activity such as analytical thought and anixiety [15]. Muse headband [16], Emotiv EPOC+ [17], and NeuroSky MindWave [18] are examples of commercial EEG headsets. fNIRS measures brain activity by monitoring changes in blood oxygenation and blood volume in the cortex, using near-infrared light [19]. This provides insights into which areas of the brain are more active, reflecting increased neuronal activity and metabolism. Examples of commercial fNIRS wearable devices include Mendi [20] and Kernel [21]. PPG is an optical technique that measures the changes in blood volume in the microvascular bed of tissue [22]. While not a direct measure of brain activity, PPG can be used alongside EEG or fNIRS to investigate the interaction between physiological markers related to brain function, such as heart rate variability [23].” (page 2, paragraph 2)
Comments 3: Although the abstract emphasizes EEG, fNIRS, and PPG devices, literature is only found on EEG-based devices, which needa thorough discussion. Technical challenges could be addressed briefly.
Response 3: We thank you for your comment. There are currently no published papers found on self-applied, remote fNIRS and PPG (in the context of neuroimaging). We mentioned fNIRS and PPG in the abstract due to the fact that these methods are widely used as portable neuroimaging modalities. We have highlighted this lack of existing literature on fNIRS and PPG in the discussion section of the paper.
“Another opportunity exists to expand remote monitoring to fNIRS and PPG, which were absent from the current literature. fNIRS allows researchers to study how activity in specific brain regions is related to various health conditions and behaviors[62]. Moreover, combining fNIRS with EEG provides a multimodal approach that capitalizes on the strengths of each technique [63]. EEG can detect changes in brain electrical activity very quickly (high temporal resolution), while fNIRS provides better spatial information about which specific brain regions are active (high spatial resolution). By combining these two techniques, researchers can track both the timing of brain activity and the location of that activity at the same time with good precision. This gives a more complete picture of how the brain's electrical signals and blood flow work together during thinking, behavior, and brain disorders [64]. PPG is also recognized to complement EEG data in monitoring different health conditions such as mental health [65]. Some wearable EEG headbands, such as Muse, and other wearable devices already incorporate PPG sensors, which is a big facilitator for tracking and analyzing this data.” (page 15, paragraph 2)
Additionally, we have expanded upon technical challenges in the discussion section of our manuscript.
“Ensuring reliable signal quality emerges as a critical challenge for remote neuroimaging with wearable EEG devices. Consumer-grade headsets often suffer from limited channel coverage, increased susceptibility to motion artifacts, and vulnerability to environmental noise interference, compromising data fidelity compared to clinical-grade systems. Rigorous signal processing techniques, such as advanced artifact removal algorithms and adaptive filtering methods, are imperative to mitigate these issues and extract accurate neural signals in real-world settings [70]. Moreover, robust validation strategies are necessitated to establish statistically significant agreement between remotely acquired data and clinical neuroimaging standards, particularly for diagnostic applications [71,72]. Overcoming these signal quality challenges through innovative hardware designs, optimized electrode configurations, and sophisticated computational methods remains a key technical hurdle to unlocking the full potential of wearable EEG devices for continuous, ecological brain monitoring outside traditional laboratory environments.” (page 15, paragraph 6)
Comments 4: I believe the article could significantly benefit from the inclusion of more sources and in-depth discussions on the findings, rather than merely listing references.
Response 4: Thank you for your comment. We have re-evaluated the eligibility for research articles and decided to include 2 new articles into the review. The studies at least partially fit the eligibility criteria (self-applied, home monitoring). Kindly find the newly cited papers listed below:
1. Levendowski DJ, Ferini-Strambi L, Gamaldo C, Cetel M, Rosenberg R, Westbrook PR. The Accuracy, Night-to-Night Variability, and Stability of Frontopolar Sleep Electroencephalography Biomarkers. J Clin Sleep Med. 2017 Jun 15;13(6):791-803. doi: 10.5664/jcsm.6618 2. Hunkin, H., King, D. L., & Zajac, I. T. (2020). EEG neurofeedback during focused attention meditation: Effects on state mindfulness and Meditation Experiences. Mindfulness, 12(4), 841–851. https://doi.org/10.1007/s12671-020-01541-0
Moreover, we have extended our discussion of the findings as shown below:
“The reviewed applications of remote EEG tracking in health monitoring focused on measuring the EEG frequency bands such as alpha, beta, theta, and delta to assess health conditions. Moreover, proprietary EEG processing algorithms were utilized in some studies to detect sleep stages and shown to be of similar accuracy to lab PSG. Similarly, remote neurofeedback interventions provided real-time feedback during the training by tracking changes in specific frequency bands associated with the targeted conditions [61]. Separating the EEG signal into different frequency bands is referred to as band analysis. Other approaches of analyzing EEG data include event-related potentials (analyzing the brain's response to specific stimuli or events) and connectivity analysis (examining the functional and effective connectivity between different brain regions) [14]. An opportunity exists to leverage these approaches on data from mobile EEG devices, allowing remote monitoring of wide verity of neurological conditions, cognitive processes, and brain networks.”(page 14, paragraph 7) |
|
|

Reviewer 2 Report
Comments and Suggestions for Authors
Despite the end result of this review is quite narrow dropping from >1600 initial papers to a 21 selected, I believe that the topic is appealing to the reader of this journal hence I would like to recommend publication.
Perhaps can be useful to sligthly enlarge the scope of the review to include at least the (Neuroimaging conducted under technicians’ supervision at home (n = 9)) if the device was indeed a simple commercial wearable headset...
Author Response
|
Response to Reviewer 2 Comments
|
|
Comments 1: Despite the end result of this review is quite narrow dropping from >1600 initial papers to a 21 selected, I believe that the topic is appealing to the reader of this journal hence I would like to recommend publication. Perhaps can be useful to sligthly enlarge the scope of the review to include at least the (Neuroimaging conducted under technicians’ supervision at home (n = 9)) if the device was indeed a simple commercial wearable headset...
|
|
Response 1: Thank you for your comment and for highlighting the importance of broadening the scope of our review. We reassessed the relevance of the nine studies involving neuroimaging conducted under technicians’ supervision at home. Kindly find a list of the studies bellow:
1. Kang, C., An, S., Kim, H. J., Devi, M., Cho, A., Hwang, S., & Lee, H. W. (2023). Age-integrated artificial intelligence framework for sleep stage classification and obstructive sleep apnea screening. Frontiers in Neuroscience, 17. https://doi.org/10.3389/fnins.2023.1059186 2. Elmali AD, Begley K, Chester H, Cooper J, Moreira C, Sharma S, Whelan A, Leschziner G, Richardson MP, Stern W, Koutroumanidis M. Evaluation of absences and myoclonic seizures in adults with genetic (idiopathic) generalized epilepsy: a comparison between self-evaluation and objective evaluation based on home video-EEG telemetry. Epileptic Disord. 2021 Oct 1;23(5):719-732. doi: 10.1684/epd.2021.1325. 3. Kataoka H, Takatani T, Sugie K. Two-Channel Portable Biopotential Recording System Can Detect REM Sleep Behavioral Disorder: Validation Study with a Comparison of Polysomnography. Parkinsons Dis. 2022 Feb 24;2022:1888682. doi: 10.1155/2022/1888682. 4. Shustak S, Inzelberg L, Steinberg S, Rand D, David Pur M, Hillel I, Katzav S, Fahoum F, De Vos M, Mirelman A, Hanein Y. Home monitoring of sleep with a temporary-tattoo EEG, EOG and EMG electrode array: a feasibility study. J Neural Eng. 2019 Apr;16(2):026024. doi: 10.1088/1741-2552/aafa05. Epub 2018 Dec 19. 5. Younes M, Younes M, Giannouli E. Accuracy of Automatic Polysomnography Scoring Using Frontal Electrodes. J Clin Sleep Med. 2016 May 15;12(5):735-46. doi: 10.5664/jcsm.5808 6. Carpentier N, Jonas J, Schaff JL, Koessler L, Maillard L, Vespignani H. The feasibility of home polysomnographic recordings prescribed for sleep-related neurological disorders: a prospective observational study. Neurophysiol Clin. 2014 Sep;44(3):251-5. doi: 10.1016/j.neucli.2014.08.005. Epub 2014 Aug 23 7. Urfer-Maurer N, Brand S, Holsboer-Trachsler E, Grob A, Weber P, Lemola S. Correspondence of maternal and paternal perception of school-aged children's sleep with in-home sleep-electroencephalography and diary-reports of children's sleep. Sleep Med. 2018 Aug;48:180-186. doi: 10.1016/j.sleep.2018.05.006. 8. Levendowski DJ, Ferini-Strambi L, Gamaldo C, Cetel M, Rosenberg R, Westbrook PR. The Accuracy, Night-to-Night Variability, and Stability of Frontopolar Sleep Electroencephalography Biomarkers. J Clin Sleep Med. 2017 Jun 15;13(6):791-803. doi: 10.5664/jcsm.6618 9. Hunkin, H., King, D. L., & Zajac, I. T. (2020). EEG neurofeedback during focused attention meditation: Effects on state mindfulness and Meditation Experiences. Mindfulness, 12(4), 841–851. https://doi.org/10.1007/s12671-020-01541-0
Upon a re-evaluation excluded papers, we have found that only papers number 8 and 9 fit the criteria of using commercial wearable headsets, as suggested by the reviewer. The rest of the papers employed devices specialized medical devices designed for clinical use, and thus would not fit the scope of this review. The total number of papers included in the review is 23 papers and the rest of the information has been updated in the manuscript.
“In [55], 63 participants were instructed to self-apply an EEG headset called Sleep Profiler at home for 2 consecutive nights. Sleep Profiler acquires EEG reading from 3 channels: AF7-AF8, AF7-Fpz, and AF8-Fpz. The EEG data was then processed to calculate sleep stages and sleep biomarkers for each night (such as sleep time, latency, cortical arousal index, and spindle durations). Certain sleep biomarkers were associated with self-reported medications, showing enhanced spindle activity and elevated sigma and beta power in sleep stages N2 and N3 among antidepressant users (P < .01). In contrast, individuals on antihypertensive medications experienced a decrease in deep sleep (N3), an increase in lighter sleep (N2), and lower levels of delta, theta, alpha, and sigma power during these stages (P < .001).” (page 10, paragraph 4)
“In [38], a crossover trial with 68 adults evaluated the effects of auditory EEG neurofeedback from Muse headband on state mindfulness and meditation experiences during focused attention meditation. Receiving auditory feedback was associated with a near-significant 15% increase in correct breath counting, suggesting improved state mindfulness. Feedback also led to significantly lower device-measured mind wandering, but negligible effects on recoveries from mind wandering episodes. Qualitative analysis revealed using feedback via Muse at home to was seen as both helpful for guiding meditation and unhelpful due to being stressful, distracting, or incongruent with subjective experience. Moreover, the study reported high adherence rate of 81% to using Muse at home over the period of 14 days.” (page 10, paragraph 7)
|
|
“In [55], the first stage of the study consisted of simultaneous in-lab PSG and EEG recording using Sleep Profiler (n=47). The PSG data was reviewed by 5 PSG technicians independently to label the sleep stages, while the EEG recordings were processed automatically. The sleep stages chosen by the majority of the PSG technicians were then compared to the automatically chosen sleep stage from the EEG data. Results has shown sleep stages drawn from EEG data had accuracy higher than 80% across different sleep sages except for N1 sleep stage (32% accuracy). Different autoscoring methods also had lower differences in the percentage of time N1 and N3 stages where selected compared with PSG technicians.” (page 14, paragraph 4)
|
